# A Bayesian Data Augmentation Approach for Learning Deep Models

**Toan Tran**[1]**, Trung Pham**[1]**, Gustavo Carneiro**[1]**, Lyle Palmer**[2] **and Ian Reid**[1]
[1]School of Computer Science, [2]School of Public Health
The University of Adelaide, Australia
`{toan.m.tran, trung.pham, gustavo.carneiro,`
`lyle.palmer, ian.reid} @adelaide.edu.au`

## Abstract

Data augmentation is an essential part of the training process applied to deep learning models. The motivation is that a robust training process for deep learning models depends on large annotated datasets, which are expensive to be acquired, stored and processed. Therefore a reasonable alternative is to be able to automatically generate new annotated training samples using a process known as data augmentation. The dominant data augmentation approach in the field assumes that new training samples can be obtained via random geometric or appearance transformations applied to annotated training samples, but this is a strong assumption because it is unclear if this is a reliable generative model for producing new training samples. In this paper, we provide a novel Bayesian formulation to data augmentation, where new annotated training points are treated as missing variables and generated based on the distribution learned from the training set. For learning, we introduce a theoretically sound algorithm — generalised Monte Carlo expectation maximisation, and demonstrate one possible implementation via an extension of the Generative Adversarial Network (GAN). Classification results on MNIST, CIFAR-10 and CIFAR-100 show the better performance of our proposed method compared to the current dominant data augmentation approach mentioned above — the results also show that our approach produces better classification results than similar GAN models.

## 1 Introduction

Deep learning has become the "backbone" of several state-of-the-art visual object classification [19, 14, 25, 27], speech recognition [17, 12, 6], and natural language processing [4, 5, 31] systems. One of the many reasons that explains the success of deep learning models is that their large capacity allows for the modeling of complex, high dimensional data patterns. The large capacity allowed by deep learning is enabled by millions of parameters estimated within annotated training sets, where generalization tends to improve with the size of these training sets. One way of acquiring large annotated training sets is via the manual (or "hand") labeling of training samples by human experts — a difficult and sometimes subjective task that is expensive and prone to mistakes. Another way of producing such large training sets is to artificially enlarge existing training datasets — a process that is commonly known in computer science as data augmentation (DA).

In computer vision applications, DA has been predominantly developed with the application of simple geometric and appearance transformations on existing annotated training samples in order to generate new training samples, where the transformation parameters are sampled with additive Gaussian or uniform noise. For instance, for ImageNet classification [8], new training images can be generated by applying random rotations, translations or color perturbations to the annotated images [19]. Such a DA process based on "label-preserving" transformations assumes that the noise model over these

transformation spaces can represent with fidelity the processes that have produced the labelled images. This is a strong assumption that to the best of our knowledge has not been properly tested. In fact, this commonly used DA process is known as "poor man's" data augmentation (PMDA) [28] in the statistical learning community because new synthetic samples are generated from a distribution estimated only once at the beginning of the training process.

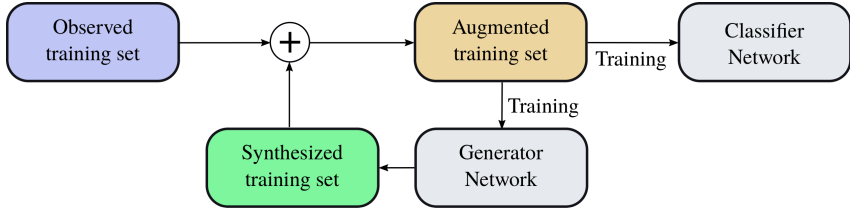

Figure 1: An overview of our Bayesian data augmentation algorithm for learning deep models. In this analytic framework, the generator and classifier networks are jointly learned, and the synthesized training set is continuously updated as the training progresses.

In the current manuscript, we propose a novel Bayesian DA approach for training deep learning models. In particular, we treat synthetic data points as instances of a random latent variable, which are drawn from a distribution learned from the given annotated training set. Effectively, rather than generating new synthetic training data prior to the training process using pre-defined transformation spaces and noise models, our approach generates new training data as the training progresses using samples obtained from an iteratively learned training data distribution. Fig. 1 shows an overview of our proposed data augmentation algorithm.

The development of our approach is inspired by DA using latent variables proposed by the statistical learning community [29], where the motivation is to introduce latent variables to facilitate the computation of posterior distributions. However, directly applying this idea to deep learning is challenging because sampling millions of network parameters is computationally difficult. By replacing the estimation of the posterior distribution by the estimation of the maximum a posteriori (MAP) probability, one can employ the Expectation Maximization (EM) algorithm, if the maximisation of such augmented posteriors is feasible. Unfortunately, this is not the case for deep learning models, where the posterior maximisation cannot reliably produce a global optimum. An additional challenge for deep learning models is that it is nontrivial to compute the expected value of the network parameters given the current estimate of the network parameters and the augmented data.

In order to address such challenges, we propose a novel Bayesian DA algorithm, called Generalized Monte Carlo Expectation Maximization (GMCEM), which jointly augments the training data and optimises the network parameters. Our algorithm runs iteratively, where at each iteration we sample new synthetic training points and use Monte Carlo to estimate the expected value of the network parameters given the previous estimate. Then, the parameter values are updated with stochastic gradient decent (SGD). We show that the augmented learning loss function is actually equivalent to the expected value of the network parameters, and that therefore we can guarantee weak convergence. Moreover, our method depends on the definition of predictive distributions over the latent variables, but the design of such distributions is hard because they need to be sufficiently expressive to model high-dimensional data, such as images. We address this challenge by leveraging the recent advances reached by deep generative models [11], where data distributions are implicitly represented via deep neural networks whose parameters are learned from annotated data.

We demonstrate our Bayesian DA algorithm in the training of deep learning classification models [15, 16]. Our proposed algorithm is realised by extending a generative adversarial network (GAN) model [11, 22, 24] with a data generation model and two discriminative models (one to discriminate between real and fake images and another to discriminate between the dataset classes). One important contribution of our approach is the fact that the modularity of our method allows us to test different models for the generative and discriminative models – in particular, we are able to test several recently proposed deep learning models [15, 16] for the dataset class classification. Experiments on MNIST, CIFAR-10 and CIFAR-100 datasets show the better classification performance of our proposed method compared to the current dominant DA approach.

# 2 Related Work

## 2.1 Data Augmentation

Data augmentation (DA) has become an essential step in training deep learning models, where the goal is to enlarge the training sets to avoid over-fitting. DA has also been explored by the statistical learning community [29, 7] for calculating posterior distributions via the introduction of latent variables. Such DA techniques are useful in cases where the likelihood (or posterior) density functions are hard to maximize or sample, but the augmented density functions are easier to work. An important caveat is that in statistical learning, latent variables may not lie in the same space of the observed data, but in deep learning, the latent variables representing the synthesized training samples belong to the same space as the observed data.

Synthesizing new training samples from the original training samples is a widely used DA method for training deep learning models [30, 26, 19]. The usual idea is to apply either additive Gaussian or uniform noise over pre-determined families of transformations to generate new synthetic training samples from the original annotated training samples. For example, Yaeger et al. [30] proposed the "stroke warping" technique for word recognition, which adds small changes in skew, rotation, and scaling into the original word images. Simard et al. [26] used a related approach for visual document analysis. Similarly, Krizhevsky et al. [19] used horizontal reflections and color perturbations for image classification. Hauberg et al. [13] proposed a manifold learning approach that is run once before the classifier training begins, where this manifold describes the geometric transformations present in the training set.

Nevertheless, the DA approaches presented above have several limitations. First, it is unclear how to generate diverse data samples. As pointed out by Fawzi et al. [10], the transformations should be "sufficiently small" so that the ground truth labels are preserved. In other words, these methods implicitly assume a small scale noise model over a pre-determined "transformation space" of the training samples. Such an assumption is likely too restrictive and has not been tested properly. Moreover, these DA mechanisms do not adapt with the progress of the learning process— instead, the augmented data are generated only once and prior to the training process. This is, in fact, analogous to the Poor Man's Data Augmentation (PMDA) [28] algorithm in statistical learning as it is non-iterative. In contrast, our Bayesian DA algorithm iteratively generates novel training samples as the training progresses, and the "generator" is adaptively learned. This is crucial because we do not make a noise model assumption over pre-determined transformation spaces to generate new synthetic training samples.

## 2.2 Deep Generative Models

Deep learning has been widely applied in training discriminative models with great success, but the progress in learning generative models has proven to be more difficult. One noteworthy work in training deep generative models is the Generative Adversarial Networks (GAN) proposed by Goodfellow et al. [11], which, once trained, can be used to sample synthetic images. GAN consists of one generator and one discriminator, both represented by deep learning models. In "adversarial training", the generator and discriminator play a "two-player minimax game", in which the generator tries to fool the discriminator by rendering images as similar as possible to the real images, and the discriminator tries to distinguish the real and fake ones. Nonetheless, the synthetic images generated by GAN are of low quality when trained on the datasets with high variability [9]. Variants of GAN have been proposed to improve the quality of the synthetic images [22, 3, 23, 24]. For instance, conditional GAN [22] improves the original GAN by making the generator conditioned on the class labels. Auxiliary classifier GAN (AC-GAN) [24] additionally forces the discriminator to classify both real-or-fake sources as well as the class labels of the input samples. These two works have shown significant improvement over the original GAN in generating photo-realistic images. So far these generative models mainly aim at generating samples of high-quality, high-resolution photo-realistic images. In contrast, we explore generative models (in the form of GANs) in our proposed Bayesian DA algorithm for improving classification models.

# 3 Data Augmentation Algorithm in Deep Learning

## 3.1 Bayesian Neural Networks

Our goal is to estimate the parameters of a deep learning model using an annotated training set denoted by $\mathcal{Y} = \{\mathbf{y}_n\}_{n=1}^N$, where $\mathbf{y} = (t, \mathbf{x})$, with annotations $t \in \{1, ..., K\}$ ($K = \#$ Classes), and data samples represented by $\mathbf{x} \in \mathbb{R}^D$. Denoting the model parameters by $\theta$, the training process is defined by the following optimisation problem:

$$\theta^* = \arg\max_\theta \log p(\theta|\mathbf{y}), \tag{1}$$

where the observed posterior $p(\theta|\mathbf{y}) = p(\theta|t, \mathbf{x}) \propto p(t|\mathbf{x}, \theta)p(\mathbf{x}|\theta)p(\theta)$.

Assuming that the data samples in $\mathcal{Y}$ are conditionally independent, the cost function that maximises (1) is defined as [1]:

$$\log p(\theta|\mathbf{y}) \approx \log p(\theta) + \frac{1}{N}\sum_{n=1}^N (\log p(t_n|\mathbf{x}_n, \theta) + \log p(\mathbf{x}_n|\theta)), \tag{2}$$

where $p(\theta)$ denotes a prior on the distribution of the deep learning model parameters, $p(t_n|\mathbf{x}_n, \theta)$ represents the conditional likelihood of label $t_n$, and $p(\mathbf{x}_n|\theta)$ is the likelihood of the data $\mathbf{x}$.

In general, the training process to estimate the model parameters $\theta$ tends to over-fit the training set $\mathcal{Y}$ given the large dimensionality of $\theta$ and the fact that $\mathcal{Y}$ does not have a sufficiently large amount of training samples. One of the main approaches designed to circumvent this over-fitting issue is the automated generation of synthetic training samples — a process known as data augmentation (DA). In this work, we propose a novel Bayesian approach to augment the training set, targeting a more robust training process.

## 3.2 Data Augmentation using Latent Variable Methods

The DA principle is to increase the observed training data $\mathbf{y}$ using a latent variable $\mathbf{z}$ that represents the synthesised data, so that the augmented posterior $p(\theta|\mathbf{y}, \mathbf{z})$ can be easily estimated [28], leading to a more robust estimation of $p(\theta|\mathbf{y})$. The latent variable is defined by $\mathbf{z} = (t^a, \mathbf{x}^a)$, where $\mathbf{x}^a \in \mathbb{R}^D$ refers to a synthesized data point, and $t^a \in \{1, ..., K\}$ denotes the associated label.

The most commonly chosen optimization method in these types of training processes involving a latent variable is the expectation-maximisation (EM) algorithm [7]. In EM, let $\theta^i$ denote the estimated parameters of the model of $p(\theta|\mathbf{y})$ at iteration $i$, and $p(\mathbf{z}|\theta^i, \mathbf{y})$ represents the conditional predictive distribution of $\mathbf{z}$. Then, the E-step computes the expectation of $\log p(\theta|\mathbf{y}, \mathbf{z})$ with respect to $p(\mathbf{z}|\theta^i, \mathbf{y})$, as follows:

$$Q(\theta, \theta^i) = \mathbb{E}_{p(\mathbf{z}|\theta^i, \mathbf{y})} \log p(\theta|\mathbf{y}, \mathbf{z}) = \int_{\mathbf{z}} \log p(\theta|\mathbf{y}, \mathbf{z}) p(\mathbf{z}|\theta^i, \mathbf{y}) d\mathbf{z}. \tag{3}$$

The parameter estimation at the next iteration, $\theta^{i+1}$, is then obtained at the M-step by maximizing the $Q$ function:

$$\theta^{i+1} = \arg\max_\theta Q(\theta, \theta^i). \tag{4}$$

The algorithm iterates until $||\theta^{i+1} - \theta^i||$ is sufficiently small, and the optimal $\theta^*$ is selected from the last iteration. The EM algorithm guarantees that the sequence $\{\theta^i\}_{i=1,2,...}$ converges to a stationary point of $p(\theta|\mathbf{y})$ [7, 28], given that the expectation in (3) and the maximization in (4) can be computed exactly. In the convergence proof [7, 28], it is assumed that $\theta^i$ converges to $\theta^*$ as the number of iterations $i$ increases, then the proof consists of showing that $\theta^*$ is a critical point of $p(\theta|\mathbf{y})$.

However, in practice, either the E-step or M-step or both can be difficult to compute exactly, especially when working with deep learning models. In such cases, we need to rely on approximation methods. For instance, Monte Carlo sampling method can approximate the integration in (3) (the E-step). This technique is known as Monte Carlo EM (MCEM) algorithm [28]. Furthermore, when the estimation of the global maximiser of $Q(\theta, \theta^i)$ in (4) is difficult, Dempster et al. [7] proposed the Generalized EM (GEM) algorithm, which relaxes this requirement with the estimation of $\theta^{i+1}$, where $Q(\theta^{i+1}, \theta^i) > Q(\theta^i, \theta^i)$. The GEM algorithm is proven to have weak convergence [28], by showing that $p(\theta^{i+1}|\mathbf{y}) > p(\theta^i|\mathbf{y})$, given that $Q(\theta^{i+1}, \theta^i) > Q(\theta^i, \theta^i)$.

### 3.3 Generalized Monte Carlo EM Algorithm

With the latent variable $\mathbf{z}$, the augmented posterior $p(\theta|\mathbf{y}, \mathbf{z})$ becomes:

$$p(\theta|\mathbf{y}, \mathbf{z}) = \frac{p(\mathbf{y}, \mathbf{z}, \theta)}{p(\mathbf{y}, \mathbf{z})} = \frac{p(\mathbf{z}|\mathbf{y}, \theta)p(\theta|\mathbf{y})p(\mathbf{y})}{p(\mathbf{z}|\mathbf{y})p(\mathbf{y})} = \frac{p(\mathbf{z}|\mathbf{y}, \theta)p(\theta|\mathbf{y})}{p(\mathbf{z}|\mathbf{y})}, \quad (5)$$

where the E-step is represented by the following Monte-Carlo estimation of $Q(\theta, \theta^i)$:

$$\hat{Q}(\theta, \theta^i) = \frac{1}{M} \sum_{m=1}^{M} \log p(\theta|\mathbf{y}, \mathbf{z}_m) = \log p(\theta|\mathbf{y}) + \frac{1}{M} \sum_{m=1}^{M} (\log p(\mathbf{z}_m|\mathbf{y}, \theta) - \log p(\mathbf{z_m}|\mathbf{y})), \quad (6)$$

where $\mathbf{z}_m \sim p(\mathbf{z}|\mathbf{y}, \theta^i)$, for $m \in \{1, ..., M\}$. In (6), if the label $t_m^a$ of the $m^{th}$ synthesized sample $\mathbf{z_m}$ is known, then $\mathbf{x}_m^a$ can be sampled from the distribution $p(\mathbf{x}_m^a|\theta, \mathbf{y}, t_m^a)$. Hence, the conditional distribution $p(\mathbf{z}|\mathbf{y}, \theta)$ can be decomposed as:

$$p(\mathbf{z}|\mathbf{y}, \theta) = p(t^a, \mathbf{x}^a|\mathbf{y}, \theta) = p(t^a|\mathbf{x}^a, \mathbf{y}, \theta)p(\mathbf{x}^a|\mathbf{y}, \theta), \quad (7)$$

where $(t^a, \mathbf{x}^a)$ are conditionally independent of $\mathbf{y}$ given that all the information from the training set $\mathbf{y}$ is summarized in $\theta$ — this means that $p(t^a|\mathbf{x}^a, \mathbf{y}, \theta) = p(t^a|\mathbf{x}^a, \theta)$, and $p(\mathbf{x}^a|\mathbf{y}, \theta) = p(\mathbf{x}^a|\theta)$.

The maximization of $\hat{Q}(\theta, \theta^i)$ with respect to $\theta$ for the M-step is re-formulated by first removing all terms that are independent of $\theta$, which allows us to reach the following derivation (making the same assumption as in (2)):

$$\hat{Q}(\theta, \theta^i) = \log p(\theta) + \frac{1}{N} \sum_{n=1}^{N} (\log p(t_n|\mathbf{x}_n, \theta) + \log p(\mathbf{x}_n|\theta)) + \frac{1}{M} \sum_{m=1}^{M} \log p(\mathbf{z}_m|\mathbf{y}, \theta) \quad (8)$$

$$= \log p(\theta) + \frac{1}{N} \sum_{n=1}^{N} (\log p(t_n|\mathbf{x}_n, \theta) + \log p(\mathbf{x}_n|\theta)) + \frac{1}{M} \sum_{m=1}^{M} (\log p(t_m^a|\mathbf{x}_m^a, \theta) + \log p(\mathbf{x}_m^a|\theta)).$$

Given that there is no analytical solution for the optimization in (8), we follow the same strategy employed in the GEM algorithm, where we estimate $\theta^{i+1}$ so that $\hat{Q}(\theta^{i+1}, \theta^i) > \hat{Q}(\theta^i, \theta^i)$.

As the function $\hat{Q}(\cdot, \theta^i)$ is differentiable, we can find such $\theta^{i+1}$ by running one step of gradient decent. It can be seen that our proposed optimization consists of a marriage between MCEM and GEM algorithms, which we name: Generalized Monte Carlo EM (GMCEM). The weak convergence proof of GMCEM is provided by Lemma 1.

**Lemma 1.** *Assuming that $\hat{Q}(\theta^{i+1}, \theta^i) > \hat{Q}(\theta^i, \theta^i)$, which is guaranteed from (8), then the weak convergence (i.e. $p(\theta^{i+1}|\mathbf{y}) > p(\theta^i|\mathbf{y})$) will be fulfilled.*

*Proof.* Given $\hat{Q}(\theta^{i+1}, \theta^i) > \hat{Q}(\theta^i, \theta^i)$, then by taking the expectation on both sides, that is $\mathbb{E}_{\mathbf{p}(\mathbf{z}|\mathbf{y}, \theta^i)}[\hat{Q}(\theta^{i+1}, \theta^i)] > \mathbb{E}_{\mathbf{p}(\mathbf{z}|\mathbf{y}, \theta^i)}[\hat{Q}(\theta^i, \theta^i)]$, we obtain $Q(\theta^{i+1}, \theta^i) > Q(\theta^i, \theta^i)$, which is the condition for $p(\theta^{i+1}|\mathbf{y}) > p(\theta^i|\mathbf{y})$ proven from [28]. $\square$

So far, we have presented our Bayesian DA algorithm in a very general manner. The specific forms that the probability terms in (8) take in our implementation are presented in the next section.

## 4 Implementation

In general, our proposed DA algorithm can be implemented using any deep generative and classification models which have differentiable optimisation functions. This is in fact an important advantage that allows us to use the most sophisticated extant models available in the field for the implementation of our algorithm. In this section, we present a specific implementation of our approach using state-of-the-art discriminative and generative models.

## 4.1  Network Architecture

Our network architecture consists of two models: a classifier and a generator. For the classifier, modern deep convolutional neural networks [15, 16] can be used. For the generator, we select the *adversarial* generative networks (GAN) [11], which include a generative model (represented by a deconvolutional neural network) and an authenticator model (represented by a convolutional neural network). This authenticator component is mainly used for facilitating the *adversarial* training. As a result, our network consists of a classifier ($C$) with parameters $\theta_C$, a generator ($G$) with parameters $\theta_G$ and an Authenticator ($A$) with parameters $\theta_A$. Fig. 2 compares our network architecture with other variants of GAN recently proposed [11, 22, 24]. On the surface, our network appears similar to AC-GAN [24], where the only difference is the separation of the classifier network from the authenticator network. However, this crucial modularisation enables our DA algorithm to replace GANs by other generative models that may become available in the future; likewise, we can use the most sophisticated classification models for $C$. Furthermore, unlike our model, the classification subnetwork introduced in AC-GAN mainly aims for improving the quality of synthesized samples, rather than for classification tasks. Nonetheless, one can consider AC-GAN as one possible implementation of our DA algorithm. Finally, our proposed GAN model is similar to the recently proposed triplet GAN [21] [1], but it is important to emphasise that triplet GAN was proposed in order to improve the training procedure for GANs, while our model represents a particular realisation of the proposed Bayesian DA algorithm, which is the main contribution of this paper.

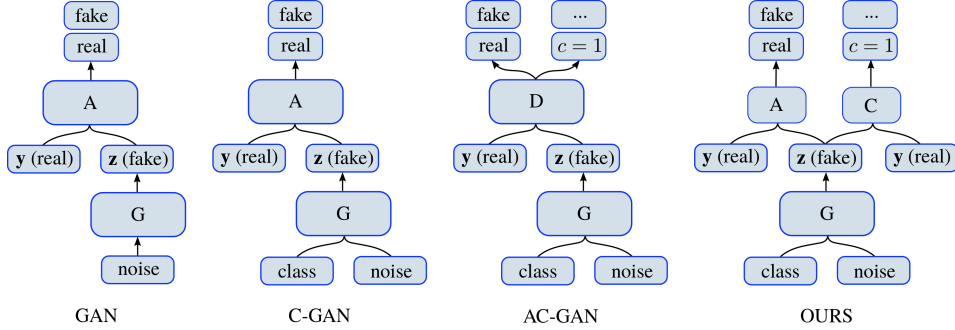

Figure 2: A comparison of different network architectures including GAN[11], C-GAN [22], AC-GAN [24] and ours. G: Generator, A: Authenticator, C: Classifier, D: Discriminator.

## 4.2  Optimization Function

Let us define $\mathbf{x} \in \mathbb{R}^D$, $\theta_C \in \mathbb{R}^C$, $\theta_A \in \mathbb{R}^A$, $\theta_G \in \mathbb{R}^G$, $u \in \mathbb{R}^{100}$, $c \in \{1, ..., K\}$, the classifier $C$, the authenticator $A$ and the generator $G$ are respectively defined by

$$f_C : \mathbb{R}^D \times \mathbb{R}^C \to [0,1]^K; \tag{9}$$

$$f_A : \mathbb{R}^D \times \mathbb{R}^A \to [0,1]^2; \tag{10}$$

$$f_G : \mathbb{R}^{100} \times \mathbb{Z}_+ \times \mathbb{R}^G \to \mathbb{R}^D. \tag{11}$$

The optimisation function used to train the classifier $C$ is defined as:

$$J_C(\theta_C) = \frac{1}{N} \sum_{n=1}^{N} l_C(t_n | \mathbf{x}_n, \theta_C) + \frac{1}{M} \sum_{m=1}^{M} l_C(t_m^a | \mathbf{x}_m^a, \theta_C), \tag{12}$$

where $l_C(t_n | \mathbf{x}_n, \theta_C) = -\log\left(\text{softmax}(f_C(t_n = c; \mathbf{x}_n, \theta_C))\right)$.

The optimisation functions for the authenticator and generator networks are defined by [11]:

$$J_{AG}(\theta_A, \theta_G) = \frac{1}{N} \sum_{n=1}^{N} l_A(\mathbf{x}_n | \theta_A) + \frac{1}{M} \sum_{m=1}^{M} l_{AG}(\mathbf{x}_m^a | \theta_A, \theta_G), \tag{13}$$

where

$$l_A(\mathbf{x}_n|\theta_A) = -\log\left(\text{softmax}(f_A(input = real, \mathbf{x}_n, \theta_A))\right); \tag{14}$$

$$l_{AG}(\mathbf{x}_m^a|\theta_A, \theta_G) = -\log\left(1 - \text{softmax}(f_A(input = real, \mathbf{x}_m^a, \theta_G, \theta_A))\right). \tag{15}$$

Following the same training procedure used to train GANs [11, 24], the optimisation is divided into two steps: the training of the discriminative part, consisting of minimising $J_C(\theta_C) + J_{AG}(\theta_A, \theta_G)$ and the training of the generative part consisting of minimising $J_C(\theta_C) - J_{AG}(\theta_A, \theta_G)$. This loss function can be linked to (8), as follows:

$$l_C(t_n|\mathbf{x}_n, \theta_C) = -\log p(t_n|\mathbf{x}_n, \theta), \tag{16}$$

$$l_C(t_m^a|\mathbf{x}_m^a, \theta_C) = -\log p(t_m^a|\mathbf{x}_m^a, \theta), \tag{17}$$

$$l_A(\mathbf{x}_n|\theta_A) = -\log p(\mathbf{x}_n|\theta), \tag{18}$$

$$l_{AG}(\mathbf{x}_m^a|\theta_A, \theta_G) = -\log p(\mathbf{x}_m^a|\theta). \tag{19}$$

### 4.3 Training

Training the network parameters $\theta$ follows the proposed GMCEM algorithm presented in Sec. 3. Accordingly, at each iteration we need to find $\theta^{i+1}$ so that $\hat{Q}(\theta^{i+1}, \theta^i) > \hat{Q}(\theta^i, \theta^i)$, which can be achieved using gradient decent. However, since the number of training and augmented samples (i.e., $N + M$) is large, evaluating the sum of the gradients over this whole set is computationally expensive. A similar issue was observed in contrastive divergence [2], where the computation of the approximate gradient required in theory an infinite number of Markov chain Monte Carlo (MCMC) cycles, but in practice, it was noted that only a few cycles were needed to provide a robust gradient approximation. Analogously, following the same principle, we propose to replace gradient decent by stochastic gradient decent (SGD), where the update from $\theta^i$ to $\theta^{i+1}$ is estimated using only a sub-set of the $M + N$ training samples. In practice, we divide the training set into batches, and the updated $\theta^{i+1}$ is obtained by running SGD through all batches (i.e, one epoch). We found that such strategy works well empirically, as shown in the experiments (Sec. 5).

## 5 Experiments

In this section, we compare our proposed Bayesian DA algorithm with the commonly used DA technique [19] (denoted as PMDA) on several image classification tasks (code available at: `https://github.com/toantm/keras-bda`). This comparison is based on experiments using the following three datasets: MNIST [20] (containing $60,000$ training and $10,000$ testing images of 10 handwritten digits), CIFAR-10[18] (consisting of $50,000$ training and $10,000$ testing images of 10 visual classes like car, dog, cat, etc.), and CIFAR-100 [18] (containing the same amount of training and testing samples as CIFAR-10, but with 100 visual classes).

The experimental results are based on the top-1 classification accuracy as a function of the amount of data augmentation used – in particular, we try the following amounts of synthesized images $M$: a) $M = N$ (i.e., $2\times$ DA), $M = 4N$ ($5\times$ DA), and $M = 9N$ ($10\times$ DA). The PMDA is based on the use of a uniform noise model over a rotation range of $[-10, 10]$ degrees, and a translation range of at most $10\%$ of the image width and height. Other transformations were tested, but these two provided the best results for PMDA on the datasets considered in this paper. We also include an experiment that does not use DA in order to illustrate the importance of DA in deep learning.

As mentioned in Sec. 1, one important contribution of our method is its ability to use arbitrary deep learning generative and classification models. For the generative model, we use the C-GAN [22] [2], and for the classification model we rely on the ResNet18 [15] and ResNetpa [16]. The architectures of the generator and authenticator networks, which are kept unchanged for all three datasets, can be found in the supplementary material. For training, we use Adadelta (with learning rate=1.0, decay rate=0.95 and epsilon=$1e-8$) for the Classifier ($C$), Adam (with learning rate $0.0002$, and exponential decay rate $0.5$) for the Generator ($G$) and SDG (with learning rate $0.01$) for the Authenticator ($A$). The noise vector used by the Generator $G$ is based on a standard Gaussian noise. In all experiments, we use training batches of size 100.

Comparison results using ResNet18 and ResNetpa networks are shown in Figures 3 and 4. First, in all cases it is clear that DA provides a significant improvement in the classification accuracy – in general,

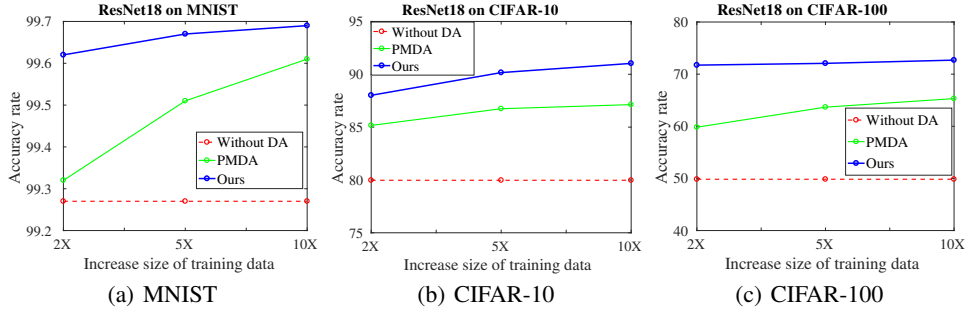

Figure 3: Performance comparison using ResNet18 [15] classifier.

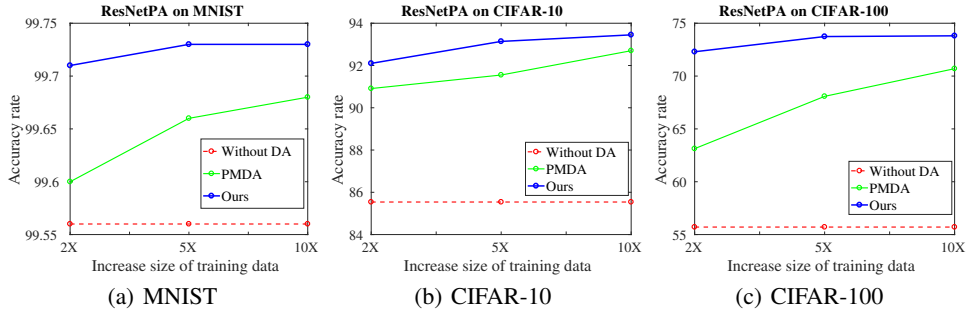

Figure 4: Performance comparison using ResNetpa [16] classifier.

larger augmented training set sizes lead to more accurate classification. More importantly, the results reveal that our Bayesian DA algorithm outperforms PMDA by a large margin in all datasets. Given the similarity between the model used by our proposed Bayesian DA algorithm (using ResNetpa [16]) and AC-GAN, it is relevant to present a comparison between these two models, which is shown in Fig. 5 – notice that our approach is far superior to AC-GAN. Finally, it is also important to show the evolution of the test classification accuracy as a function of training time – this is reported in Fig. 6. As expected, it is clear that PMDA produces better classification results at the first training stages, but after a certain amount of training, our Bayesian DA algorithm produces better results. In particular, using the ResNet18 [15] classifier, on CIFAR-100, our method is better than PMDA after two hours of training; while for MNIST, our method is better after five hours of training.

It is worth emphasizing that the main goal of the proposed Bayesian DA is to improve the training process of the classifier $C$. Nevertheless, it is also of interest to investigate the quality of the images produced by the generator $G$. In Fig. 7, we display several examples of the synthetic images produced by $G$ after the training process has converged. In general, the images look reasonably realistic, particularly the handwritten digits, where the synthesized images would be hard to generate

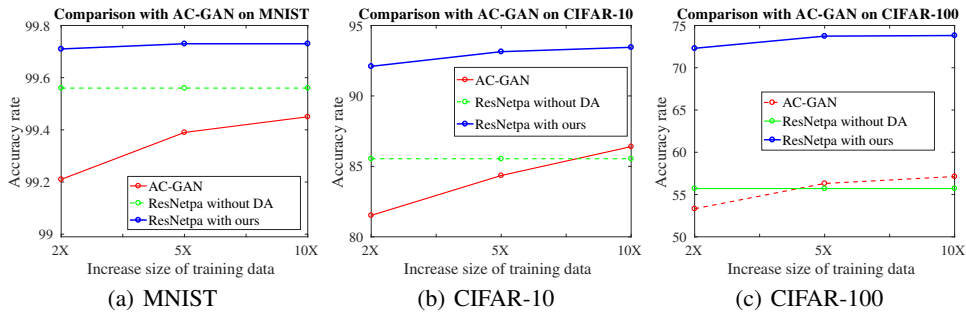

Figure 5: Performance comparison with AC-GAN using ResNetpa [16]

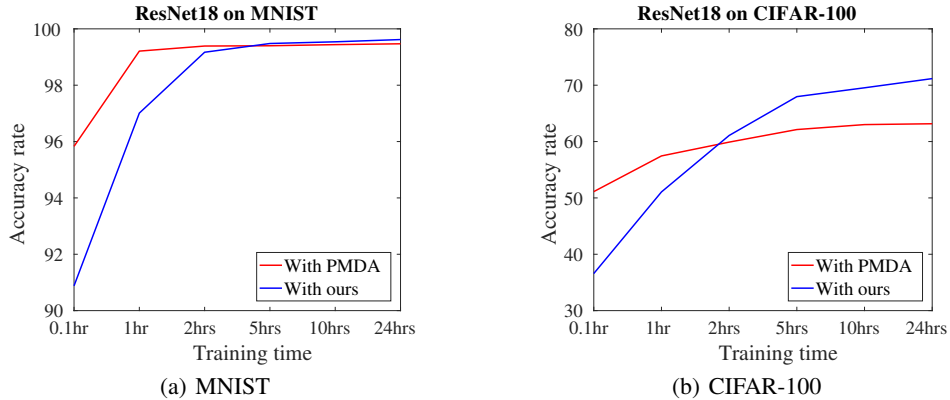

| | |
|---|---|
| (a) MNIST | (b) CIFAR-100 |

Figure 6: Classification accuracy (as a function of the training time) using PMDA and our proposed data augmentation on ResNet18 [15]

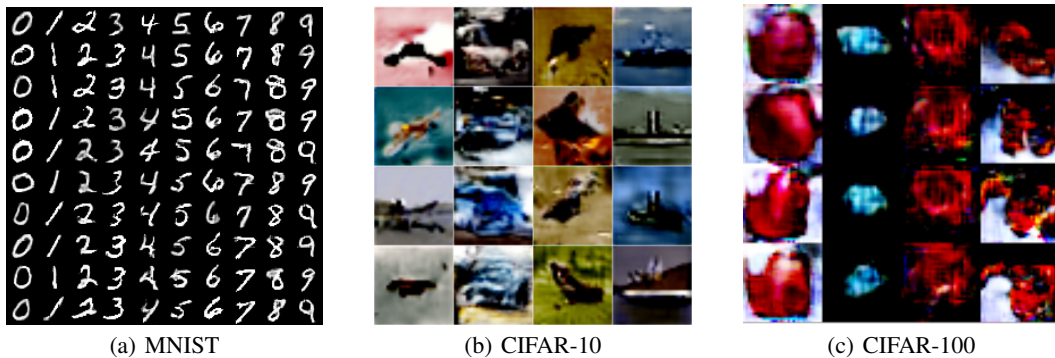

| | | |
|---|---|---|
| (a) MNIST | (b) CIFAR-10 | (c) CIFAR-100 |

Figure 7: Synthesized images generated using our model trained on MNIST (a), CIFAR-10 (b) and CIFAR-100 (c). Each column is conditioned on a class label: a) classes are $0, ..., 9$; b) classes are airplane, automobile, bird and ship; and c) classes are apple, aquarium fish, rose and lobster.

by the application of Gaussian or uniform noise on pre-determined geometric and appearance transformations.

# 6 Conclusions

In this paper we have presented a novel Bayesian DA that improves the training process of deep learning classification models. Unlike currently dominant methods that apply random transformations to the observed training samples, our method is theoretically sound; the missing data are sampled from the distribution learned from the annotated training set. However, we do not train the generator distribution independently from the training of the classification model. Instead, both models are jointly optimised based on our proposed Bayesian DA formulation that connects the classical latent variable method in statistical learning with modern deep generative models. The advantages of our data augmentation approach are validated using several image classification tasks with clear improvements over standard DA methods and also over the recently proposed AC-GAN model [24].

# Acknowledgments

TT gratefully acknowledges the support by Vietnam International Education Development (VIED). TP, GC and IR gratefully acknowledge the support of the Australian Research Council through the Centre of Excellence for Robotic Vision (project number CE140100016) and Laureate Fellowship FL130100102 to IR.

## Footnotes

[1]The triplet GAN [21] was proposed in parallel to this NIPS submission.

[2]The code was adapted from: `https://github.com/lukedeo/keras-acgan`

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
