[Reviews · NeurIPS 2017]

Reviewer 1



Data augmentation for neural networks generates synthetic examples, which are then used for additional training of a classifier. This can be achieved by label preserving transformations in a generic way or adaptive by training a generative adversarial network (GAN). The authors propose a new network architecture for this task consisting of three deep neural networks, two classifiers and one generator. The training algorithm described in the paper combines Monte Carlo expectation maximization (MCEM) and generalized expectation maximization (GEM). Experiments on standard data sets for image classification show that the new method is significantly better than data augmentation with label preserving transformations. The paper is well written. It clearly explains the proposed network architecture as well as the algorithm for training the three neural networks. However, "-log(1 - fA(fake))" in (15) is not consistent with the other cost functions and should be corrected. As the new approach is an extension of AC-GAN, a comparison with that architecture would be most interesting in order to see the effect of adding a third network directly. Therefore I recommend to add results for AC-GAN to the next version of this paper. While the results provided in the paper show that GAN methods are better than label preserving transformations or no augmentation, this does not tell if adding a third network is justified by a significant increase in performance. The comparisons provided in the author feedback show that the proposed network architecture has advantages over previous data augmentation methods. Including these results as well as the correction for (15) in the paper will improve it. However, a more systematic comparison would be even better.

Reviewer 2



The authors propose a Bayesian approach to handle Data Augmentation (DA) for learning deep models. Data augmentation, in this context, addresses the problem of data sparsity by generating more training samples to make the training process of a (heavily parameterized) Classifier Network (CN) robust. The standard Generative Model (GM) for obtaining the augmented training set is typically trained once and then new data are generated based on this GM and used to estimate the CN. The authors suggest to also learn the GM itself jointly with the CN as new augmented training data are produced - which is straightforward with a Bayesian approach. Using three different data sets and two classifiers, they demonstrate that the Bayesian DA is superior to regular DA, which in turn is superior to not having DA at all (which likely suffers from some overfitting). The authors claim in the introduction that they provide a novel Bayesian formulation of DA. While this may be true, from a Bayesian point of view their idea is quite simple, as this is what any Bayesian statistician would do. Having said that, I think this is a good paper which is clearly written. If it really is the first attempt to think about this problem in a Bayesian way it is by itself worthy of publication, but I am not sufficiently familiar with the deep learning literature to judge this. The main problem with the pure DA is that one is heavily relying of the GM to be producing correct data, while the proposed approach would allow the GM to be improved. This makes the authors' approach more challenging as both the CN and the GM need to be trained and, in top of this, both are deep models. Hence, making inference using traditional powerful Bayesian tools seems impossible and the authors propose an algorithm which is in spirit with the EM-algorithm to learn the "latents" (here the artificial data) conditional on model parameters and vice versa. Hence my main concern regards the optimization and its robustness to different choices of tuning parameters. For example, Wilson et al. (2017) show that adaptive step-size choices (such as ADAM) are prone to find poor modes in heavily parameterized models. This model is significantly more parameterized than the standard DA, and, although you generate more data, you also have the problem of training the GM. Can you comment on this? I think that Lemma 1 should be expressed more carefully, stating the assumptions made. @article{wilson2017marginal, title={The Marginal Value of Adaptive Gradient Methods in Machine Learning}, author={Wilson, Ashia C and Roelofs, Rebecca and Stern, Mitchell and Srebro, Nathan and Recht, Benjamin}, journal={arXiv preprint arXiv:1705.08292}, year={2017} }